# Cyclic Electron Flow-Coupled Proton Pumping in *Synechocystis* sp. PCC6803 Is Dependent upon NADPH Oxidation by the Soluble Isoform of Ferredoxin:NADP-Oxidoreductase

**DOI:** 10.3390/microorganisms10050855

**Published:** 2022-04-21

**Authors:** Neil T. Miller, Ghada Ajlani, Robert L. Burnap

**Affiliations:** 1Department of Microbiology and Molecular Genetics, Oklahoma State University, Stillwater, OK 74078, USA; ntmiller2@wisc.edu; 2Institute for Integrative Biology of the Cell (I2BC), Université Paris-Saclay, CNRS, CEA, 91198 Gif-sur-Yvette, France; gajlani@cea.fr

**Keywords:** cyanobacteria, cyclic electron flow, ferredoxin-NADP reductase, NDH-1, photosynthesis, proton motive force, thylakoid

## Abstract

Ferredoxin:NADP-oxidoreductase (FNR) catalyzes the reversible exchange of electrons between ferredoxin (Fd) and NADP(H). Reduction of NADP^+^ by Fd via FNR is essential in the terminal steps of photosynthetic electron transfer, as light-activated electron flow produces NADPH for CO_2_ assimilation. FNR also catalyzes the reverse reaction in photosynthetic organisms, transferring electrons from NADPH to Fd, which is important in cyanobacteria for respiration and cyclic electron flow (CEF). The cyanobacterium *Synechocystis* sp. PCC6803 possesses two isoforms of FNR, a large form attached to the phycobilisome (FNR_L_) and a small form that is soluble (FNR_S_). While both isoforms are capable of NADPH oxidation or NADP^+^ reduction, FNR_L_ is most abundant during typical growth conditions, whereas FNR_S_ accumulates under stressful conditions that require enhanced CEF. Because CEF-driven proton pumping in the light–dark transition is due to NDH-1 complex activity and they are powered by reduced Fd, CEF-driven proton pumping and the redox state of the PQ and NADP(H) pools were investigated in mutants possessing either FNR_L_ or FNR_S_. We found that the FNR_S_ isoform facilitates proton pumping in the dark–light transition, contributing more to CEF than FNR_L_. FNR_L_ is capable of providing reducing power for CEF-driven proton pumping, but only after an adaptation period to illumination. The results support that FNR_S_ is indeed associated with increased cyclic electron flow and proton pumping, which is consistent with the idea that stress conditions create a higher demand for ATP relative to NADPH.

## 1. Introduction

Cyclic electron flow (CEF) is an important mechanism in photosynthetic organisms as a means of recycling excess reductant while simultaneously driving the synthesis of ATP by the generation of proton motive force (PMF) [1,2,3]. In plants, algae, and cyanobacteria the reducing power produced by the photosynthetic light reactions is stored in the form of NADPH and Fd_red_, with the former being consumed by anabolic processes, mainly CO_2_ fixation via the Calvin–Benson–Bassham (CBB) cycle. This process is driven by linear electron flow (LEF) since electrons derived from the oxidation of water by photosystem II (PSII) follow a linear sequence of transfers through the photosynthetic electron transport chain, on through photosystem I (PSI), and ultimately terminating in the reduction of inorganic substrates for biomass production. During CEF, however, accumulated NADPH and Fd_red_ follow an alternative path that returns the energized electrons back to the membrane where they are oxidized by protein redox complexes in the electron transport chain, where depending on the pathway, they may drive the formation of PMF and thereby contribute to ATP production without net consumption of NADPH and Fd_red_. The fraction of electrons recycled through CEF relative to LEF allows the cells to adjust the output ratio of ATP/NADPH to accommodate different metabolic demands and fluctuating environmental conditions. Recently, cellular ATP concentrations were shown to modify the rate of CEF through a competitive inhibition of Fd oxidation by the NDH and FQR/PGR5 pathways in plant chloroplasts, providing a relatively simple feedback mechanism for adjusting this output ratio of phosphorylating and reducing power [4].

Although alternative cyclic paths involving different membrane complexes have been discovered, in most cases electrons are transferred to the plastoquinone (PQ) pool, thereby re-entering the electron transport chain to be further utilized to produce NADPH and Fd_red_ via PSI and subsequently recycled via CEF. Moreover, the overall CEF pathway may have many potential electron sources and contributing components. In cyanobacteria, the major routes by which electrons re-enter the PQ pool via CEF are NDH-1 and PGR5 [1,2,5,6,7]. Previously, in the dark–light transition NDH-1 was shown to be the major contributor to ∆pH formation by CEF in *Synechocystis* sp. PCC 6803, with the mutants unable to pump protons as efficiently after the loss of either only the respiratory-like NDH-1 complexes or loss of all NDH-1 complexes present [8]. In addition to this, pseudocyclic electron flow via the flavodiiron proteins, Flv1/3, may act as a way to dissipate electrons from Fd_red_/NADPH to reduce O_2_-yielding water, and is an important photoprotective mechanism, especially under conditions of fluctuating light and inorganic carbon limitation [3,9,10]. Because flavodiiron protein reduction of O_2_ occurs in the cytoplasm, the contributions to proton motive force, and hence ATP production, are likely restricted to enabling unfettered photosystem II water-oxidation activity, with the concomitant release of protons into the thylakoid lumen [3,10].

Cyclic electron flow reentering the PQ pool can be directly from the acceptor side of PSI via Fd, or indirectly from the oxidation of photosynthetically produced NADPH, or carbohydrates [11,12,13,14]. Both of these CEF inputs contribute, but with different apparent kinetics, for example, in the post-illumination rise in Chl fluorescence yield [12]. One way the redox state of the NADPH/Fd pool is maintained is by the redox activity of Ferredoxin:NADP-oxidoreductase (FNR). In many cyanobacteria, only one form of FNR is present (FNR_L_), and it attaches to the phycobilisome (PBS), poising it to function in photosynthetic two-electron reduction of NADP^+^ by Fd_red_ from the acceptor side of PSI [15,16,17,18,19]. FNR_L_ localization in proximity of the photosynthetic membrane via attachment to the PBS is thought to facilitate fast rates of LEF. Specifically, it would minimize macromolecular diffusion constraints because the interaction of PSI, Fd, and FNR allows for the rapid production of quickly diffusing NADPH, which is in high demand and undergoes fast recycling between the oxidized and reduced forms as it communicates reductant from photosynthetic electron transport to the CBB cycle and anabolic metabolism in the cytoplasm [20].

Some cyanobacteria, such as the glucose tolerant *Synechocystis* sp. PCC6803, possess two isoforms of FNR. The second isoform is soluble and smaller, and therefore is referred to as FNR_S_. It was discovered to be produced by the same gene [16] and is produced by an alternative initiation of translation, via mRNA secondary structure formation [21]. The isoform primarily observed in cyanobacteria is FNR_L_, which may have arisen due to a gene recombination event of FNR_S_ with a PBS-linker domain, eventually resulting in the genetic conversion FNR_S_ to FNR_L_ in most of these bacteria [16]. This is consistent with the observation that *Gloeobacter violaceus*, a primitive cyanobacterium, possesses only the FNR_S_ form [16]. Cyanobacteria capable of heterotrophy, on the other hand, have retained the ability to recapitulate the FNR_S_ isoform through regulated alternative translational initiation codon utilization, and it is expressed under conditions of heterotrophy, or stress conditions such as high light intensity or nutrient limitation [16], which necessitates CEF activity [1]. In deletion mutants of either FNR_L_ (FS1) or FNR_S_ (MI6), similar amounts of total FNR accumulate compared to the WT, and though a small amount of FNR is detected in the smaller band in MI6, this is either a proteolytic cleavage product or a functionally insignificant product beginning at codon 102 [16]. The fact that these organisms have kept the ability to express the FNR_S_ isoform suggests that it is an important part of their bioenergetic apparatus. Data on the kinetics of these isoforms has led to the hypothesis that FNR_S_ operates as a soluble NADPH oxidase, while FNR_L_ acts more as an NADP^+^ reductase associated with the PBS, though both isoforms are capable of performing either reaction [17]. This hypothesis is supported by the impairment of photoautotrophic growth in the mutant lacking FNR_L_ [16]. Because cyanobacterial NDH-1 complexes utilize Fd as their source of reductant [14,22] and their contribution to proton pumping can be observed by AO fluorescence [8], the coupling of electron transport and proton pumping data can be used to explore the dynamics of the relationship between these two isoforms, CEF, and proton pumping. Here, this relationship is explored by utilizing these techniques and strains lacking either FNR_S_ or FNR_L_.

## 2. Materials and Methods

Strains of the glucose-tolerant *Synechocystis* sp. PCC 6803 (hereafter, *Synechocystis*) were maintained on pH 8 BG-11 [23] with 1.5% agar supplemented with 18mM sodium thiosulfate and containing the 25 µg/mL spectinomycin for the FS1 and MI6 mutants. They were grown under ~70 µE m^−2^ s^−1^ Cool White fluorescent lighting (GE) and air levels of CO_2_. Experimental material was obtained from 100 mL cultures grown in 250 mL Erlenmeyer flasks with rotary shaking (200 rpm) under ~100 µE m^−2^ s^−1^ Cool White fluorescent lighting (GE). Cultures were harvested upon reaching an OD_750_ 0.7–1.0. The strains MI6 and FS1 were constructed as described previously [16].

Cells were harvested by centrifugation, washed with 50 mM Tricine pH 8 + 25 mM KCl (TCK) and resuspended to 5 µg/mL Chl, measured by UV-Vis spectrophotometer (Shimadzu, Kyoto, Japan) [24]. Measurements of Chl and NADPH fluorescence kinetics were acquired using the Dual PAM-100 (Walz, Effeltrich, Germany) with the 9-AA/NADPH module. For 5 min illumination-period measurements, a nominal actinic light intensity of 53 µE m^−2^ s^−1^ and wavelength of 635 nm was utilized with multiple turnover (MT, nominally 20,000 µE m^−2^ s^−1^) flashes occurring before, during, and after illumination to visualize the approximate proportion of open PSII [25]. Chlorophyll fluorescence was monitored with pulse amplitude modulated (PAM) LED excitation at 620 nm with detection at >700 nm. NADPH fluorescence was followed with PAM LED excitation at 365 nm and detection of emission within the broad blue-green band (420–580 nm) [26]. The application of an MT flash in NADPH fluorescence measurements produces an artifactual feature indicative of a very fast oxidation and a gradual increase back to steady state after the flash, but it is unclear to what extent this feature is artifactual nor is its basis understood, therefore it was omitted from the graphical representation of the data in order to focus on the known features.

Acridine orange (AO) has been used as a fluorescent dye for measuring proton pumping across a membrane in both whole cyanobacterial cells and membrane vesicles [8,27,28,29,30] and the assay protocol outlined previously in [8] was utilized with minor modifications. Cells were grown as described above and washed with TCK buffer as above for Chl fluorescence measurements. Samples were diluted to 5.9 µg/mL Chl in TCK buffer and acridine orange added to 5 µM, and were incubated while shaking gently in the dark for 20 min to allow acridine orange penetration into the cell [29]. Samples of the cells were then added to a cuvette and the cells diluted to 5 µg/mL in TCK buffer and the inhibitors applied as described previously [8]. Samples were then stirred for 5 min in the dark, and the fluorescence measured with the JTS-100 (Biologic, Seyssinet-Pariset, France), described in greater detail below. A 534/20 nm bandpass filter (Edmund Optics, Barrington, NJ, USA) was placed between the sample and the detector.

Data points in the JTS-100 were collected once per second in the dark and once per 100 ms in the light, with 600 µE m^−2^ s^−1^ actinic light applied. Cells incubating in the dark were provided with an additional 5 min dark incubation with stirring after the addition of inhibitors. Samples were then illuminated with actinic light for 15 s, and then actinic illumination was ceased. For light-adapted samples, the cells were incubated with the inhibitors for 5 min stirring in the dark, followed by 4 min of illumination with 600 µE m^−2^ s^−1^ actinic light, followed by a further 4 min stirring in the dark before measurement in the same manner described above. Samples were refreshed after each measurement as drift in the measurements was seen when using TCK buffer.

## 3. Results

### 3.1. FNR_s_ Has a Large Contribution to NDH-1 Cyclic Electron Flow

Chlorophyll fluorescence kinetics have long been used to observe electron flow through the PQ pool in cyanobacteria [31,32], and the post-illumination kinetics are especially indicative of NDH-1 driven CEF [1,12,13,14,33,34]. In Figure 1, the chlorophyll fluorescence traces for the WT, MI6 (containing only FNR_L_), and FS1 (containing only FNR_S_) are shown. During the 5 min illumination in the TCK assay buffer, WT and FS1 had similar initial fluorescence kinetics immediately upon illumination, but the oxidation of PQ, indicated by fluorescence quenching, was outpaced by reduction of the PQ pool in FS1, which potentially obscured the small peak at ~20–30 s of illumination corresponding to activation of the CBB cycle in the light (Figure 1A,E) [12]. The WT then maintained a steady state of PQ redox after the CBB cycle activation, while FS1 showed a slow reduction in the PQ pool over illumination toward what appears to be close to a steady state with a comparatively more reduced PQ pool. The MI6 mutant, while showing similar broad features as the WT, exhibited a large initial fluorescence yield which was evident in the saturating flash in the dark period and by the large amplitude of the fluorescence yield upon continuous illumination. We attribute the high fluorescence yields in MI6 to the redox poising of cells in State 1, much like mutants of the NDH-1 complex, where a relatively oxidized PQ pool is observed in the dark [35]. Upon continuous illumination there is reduction in the PQ pool upon illumination followed by steady oxidation over the course of illumination toward values nearer the WT than FS1 (Figure 1A,C,E). This consistent with MI6 operating primarily via linear electron transport, although the influence of state transitions and other quenching processes are likely to also contribute to the progress quenching after the initial S-M rise [36,37]. In contrast to either the wild-type or MI6, there is a progressive increase in fluorescence yield in FS1, indicating that the FNR_S_ isoform is highly efficient at Fd_ox_ reduction and subsequent electron transfer to the PQ pool transfer via the NDH-1 complexes [14,22]. In this context, we note that pseudocyclic electron transfer to O_2_ via the flavodiiron proteins occurs in the cytoplasm, thereby dissipating electrons from the entire system and tending oxidize the PQ pool as the electrons have no return path.

Illumination was ceased at 360 s, and the post-illumination increase in fluorescence yield is attributed to CEF. The WT and MI6 have very minimal fluorescence rises while FS1 has a large rise that appears to include both the fast and slow components. The fast components (<10 s post-illumination) are associated with direct CEF involving NADPH and reduced ferredoxin, whereas the broad slow component (~30 s post-illumination) reflects reoxidation of photosynthetically produced sugars (Figure 1A,C,E) [12]. This indicates that in the WT and MI6 under these conditions, electrons are not flooding back into the PQ pool via redox complexes, primarily NDH-1 [8,14], after illumination, or if they are, they are being removed from the PQ pool at a similar rate to that which they are entering. This is not the case in FS1, where electrons flood the PQ pool after illumination, and the input of electrons into the PQ pool outpaces the consumption of those electrons after illumination indicated by the pronounced post-illumination rise in fluorescence relative the wild type (Figure 1E). This reflects the hypothesized role of FNR_S_ in its participation in CEF [16,18].

When KCN is added to the samples, preventing cytochrome oxidase (COX) activity, stark changes are seen in the fluorescence yield. Over the course of the 5 min illumination, WT starts out with a very reduced PQ pool, which becomes more reduced over the illumination so Chl fluorescence reaches to near the F_m_ level (Figure 1B). MI6 shows a large initial reduction in PQ upon illumination, followed by the typical quenching associated with CBB cycle activation (Figure 1D), which is followed by a slight increase in fluorescence over the course of illumination, indicating a slight reduction in the PQ pool. FS1 shows the typical initial reduction followed by a brief oxidation and then consistent steady reduction of the PQ pool over the course of illumination (Figure 1F). The post-illumination fluorescence rise is similar in the WT and MI6 in terms of magnitude. This rise is further investigated in Figure 2, compiling the post-illumination period of the measurements in Figure 1B,D,F. As can be seen, FS1 has the sharpest initial rise after illumination and reaches the highest fluorescence yield. This initial rise is also seen in MI6 and the WT, but not to the magnitude seen in FS1. After this initial rise, a second rise occurs over the course of ~12–30 s after the cessation of illumination. This slower rise takes place at similar rates for all the strains in these conditions, but the peak of the fluorescence yield and its dissipation is shifted in FS1, where even a slow dissipation does not start until ~30 s after cessation of illumination versus ~20 s in WT and MI6 (Figure 1B,D,F and Figure 2). This shows that FNR_L_, the form present in MI6 and predominant in the WT, may participate in Fd_ox_ reduction for CEF, and that FNR_S_ appears to be more effective at this function than its counterpart.

To further investigate the post-illumination Chl fluorescence rise in the absence of inhibitors, 15 s illuminations on dark-adapted samples were performed. Illuminations of this length are long enough to generate a substantial pool of photosynthetic reductant as NADPH and reduced ferredoxin, yet too short for the full activation of the CBB cycle. The FS1 and WT had similar initial slopes of the post-illumination fluorescence rise; however, the FS1 had a greater magnitude, which is associated with greater CEF activity, especially via NDH-1 [13] (Figure 3). The MI6 mutant, on the other hand, had a small rise, with dissipation of the post-illumination fluorescence occurring well before either FS1 or the WT had reached their peak and begun their own dissipations (Figure 3). These results indicate that FS1 has enhanced CEF, MI6 has diminished CEF, and the WT is somewhere in between the two. Taken together, these results indicate that only having the FNR_L_ in MI6 tends to shunt the reductant to CO_2_ reduction via the CBB with minimal CEF with the resultant more efficient oxidation of the PQ pool. With the ability to express only the short form of FNR_S_ in FS1, CEF is maximized, resulting in enhanced re-reduction of the PQ pool at the expense of LEF.

### 3.2. FNR_S_ Enhances NADPH Oxidation during Illumination

NADPH fluorescence was measured concomitantly with chlorophyll fluorescence in the DualPAM-100 to observe the redox state of the soluble reductant pool during illumination. The MT flashes produce an artifact that was excluded in the graphics to emphasize the kinetics of the NADPH redox poise on in the seconds time frame. As seen in Figure 4A, the WT exhibits characteristics typical of NADPH fluorescence during an illumination of this length [12]. The rises in fluorescence indicate the reduction of NADP^+^, and the quenching of fluorescence is indicative of NADPH oxidation. After the rapid reduction of the pool upon illumination, a small oxidation event is seen before a large transient reduction associated with the production of NADPH and representative of the State-2 to State-1 transition, where light harvesting energy is focused around PSII over PSI [38,39,40,41]. This is followed by a subsequent oxidation of NADPH associated with the activation of the CBB cycle and fixation of carbon after ~30 s of illumination [12]. After illumination ceases, there is a rise in fluorescence that starts ~10 s after illumination and continues for ~15–20 s, which is associated with the oxidation of sugars [12]. The delay before the rise correlates with the oxidation of trioses seen in the post-illumination rise of Chl fluorescence in KCN-treated cells (Figure 1 and Figure 2), indicating that the delay in the rise is associated with Fd reduction before NADP^+^ reduction by catabolism begins. The MI6 mutant has similar characteristics as the WT during and post-illumination; however, the overall NADPH pool was not as strongly reduced in the light, and its post-illumination rise is slightly less dramatic in magnitude (Figure 4C). The FS1 mutant on the other hand had distinct characteristics in its NADPH redox state over the course of illumination (Figure 4E). It had an initial reduction followed by a strong oxidation of NADPH to a steady state that was deeply oxidized compared to the WT or MI6, and even dropped below the baseline of fluorescence pre-illumination (Figure 4E). The transient reduction peak is shifted to ~15 s earlier than it is in the WT or MI6, potentially indicating changes in the regulation of the redox pools. After illumination, a large and dramatic rise was seen in FS1 to approximately the level of the initial NADPH reduction peak in a biphasic manner, first occurring quickly and then rising slowly to a point before minor oxidation is observed in the dark. This occurred on approximately the same timescale as the WT or MI6, indicating the same mechanisms may be responsible for this dramatic rise, but perhaps with a greater flux of electrons through them than in either WT or MI6. This indicates that there is a strong oxidation of NADPH by FNR_S_ that is not observed when FNR_L_ is the dominant isoform.

Upon the addition of KCN, the WT almost entirely reduced its NADPH pool in the dark, as upon illumination there is little increase in fluorescence (Figure 4B). There is a transient oxidation, and then NADPH fluorescence rises slowly and reaches a steady state for the rest of illumination. Given that the PQ pool is almost entirely reduced over the course of illumination (Figure 1B), NADPH might be as reduced as it can be as well, representing an acceptor side limitation. The MI6 has similar characteristics of fluorescence as before treatment with KCN; however, the initial rise in fluorescence upon illumination is diminished (Figure 4D). It also does not show strong reduction in the dark, or even during illumination, perhaps showing NADPH oxidase activity. A decrease in fluorescence is seen after the transient peak, indicating dissipation of reductant by CO_2_ fixation. At the beginning of post-illumination NADPH fluorescence of MI6 rises to a greater magnitude than the WT but with a similar slope, and the pool continued to reduce over time. In the dark, FS1 had a similarly reduced NADPH pool to the WT, and interestingly, upon illumination and after its transient peak, had a strong oxidation of the NADPH pool, though not quite as quickly or deeply as without KCN, followed by a slow and steady rise in fluorescence which seems to continue at a similar rate after the oxidation and subsequent reduction of NADPH after termination of illumination (Figure 4F). The post-illumination rise was much smaller in magnitude than without KCN treatment, but still greater compared to the other strains (Figure 4E,F). Together with the chlorophyll fluorescence data (Figure 1), the NADPH fluorescence data indicate that FNR_S_ pulls electrons away from the NADPH pool and allows their input into the PQ pool to a greater degree than does FNR_L_. While it is difficult to compare the physiological conditions exactly, this accords with the previous findings that the NADP^+^/NADPH ratios were about 2.6 and 2.0 for the wild type and MI6, respectively. In contrast, the reported ratio was about 4.3 in FS1, indicating a much more oxidized NADPH pool [17].

### 3.3. The Presence of FNR_s_ Enhances NDH-1 Powered Proton Pumping

To investigate the ability of these strains to power CEF-driven proton pumping and determine if Fd is indeed being reduced by the strong NADPH oxidation seen in FS1 to power CEF, ∆pH formation upon actinic illumination was measured by acridine orange (AO) fluorescence quenching. This assay was performed essentially as described previously [8]; however, the buffer used included 25 mM KCl (TCK buffer) rather than 25 mM NaCl (TCN). Upon switching the buffer, the rates of AO quenching dramatically reduced from those reported in TCN buffer (Appendix A). The fast rate of quenching in the TCN buffer was shown to be Na^+^ gradient-dependent, as the addition of 50 µM monensin dissipated the gradient, reducing the rates to approximately those seen in TCK buffer, indicating that the Na^+^ gradient plays a part in light-induced PMF formation, perhaps by modulating the concentration of positive charges across the thylakoid. As discussed below, subtler changes in AO fluorescence quenching may be observed in TCK buffer, and it enhances the proportion of PMF formation driven by CEF. Because NDH-1 complexes have been shown to be the primary contributors to CEF-driven proton pumping in the dark–light transition in cells treated with DCMU [8], and their primary reductant source is reduced Fd [14], the action of FNR_S_ as an NADPH oxidizer should be most clearly seen when PSII is inhibited. While the action of an alternative CEF pathway such as PGR5 is unable to be excluded in its activity here, based on previous data it is likely that in these conditions the ∆pH formation is due to CEF via NDH-1 complexes [8]. The background inhibitors KCN, valinomycin, and DCCD were utilized as described previously to allow for the measurement of near-maximal rates of proton pumping [8]. In dark-adapted cells, FS1 has a significantly faster rate of quenching than the WT with only the background inhibitors, but the rates are not significantly different when DCMU is added to dark-adapted cells (*p* = 0.88) (Figure 5A,B and Table 1). The mutant MI6, lacking FNR_S_, was not significantly different from the WT with only the background inhibitors in dark-adapted cells (*p* = 0.35); however, it was significantly slower than either the WT or FS1 when DCMU was added (Figure 5C and Table 1). The ability for CEF to compensate for the loss of LEF in MI6 is greatly reduced from that of the WT or FS1, operating at ~27% of the rate of the no-DCMU control, while WT and FS1 operate CEF-driven proton pumping at ~50% and 39% of the rate, respectively. These data indicate that in dark-adapted cells FNR_S_ is a better supplier of Fd_red_ for NDH-1-driven proton pumping than FNR_L_ is, and that the WT can match the rate of FS1 due to its limited presence of FNR_S_.

Upon repeated illumination of a sample in TCK buffer, increasing rates were seen in subsequent samples (not shown). This was hypothesized to be due to an increasing reductant pool feeding CEF proton pumping and allowing it to carry forward faster. To test this, cells were treated with the background inhibitors and DCMU/CCCP where appropriate, pre-illuminated for 2 min at 600 µE, let to sit in the dark for 4 min with stirring, and then subsequently measured for typical 15 s illuminations. As seen in Figure 6 and Table 1, pre-illumination of the cells resulted in increases in rates of proton pumping, especially in samples with DCMU. The proton-pumping rate upon DCMU treatment in light-adapted cells is nearly twice the rate of dark-adapted cells, while FS1 is ~2.7× as fast vs. dark-adapted cells. The MI6 increased in rates as well, essentially matching those of WT and FS1 with only the background inhibitors (Figure 6 and Table 1). FS1 had the fastest proton-pumping rate with DCMU added, indicating that FNR_S_ is important for driving proton pumping via CEF (Figure 6C and Table 1). By treating the cells with illumination prior to measurement, the ability of the cells to compensate for the loss of LEF by DCMU treatment increased in all cases. The WT was able to achieve ~76% the light-adapted control rate when treated with DCMU, while it could only achieve ~50% the rate when dark-adapted (Figure 6A and Table 1). When light-treated, MI6 was able to achieve a rate of ~50% the control when treated with DCMU, while it could only achieve ~27% when dark-adapted (Figure 6B and Table 1). FS1 was able to achieve ~89% the rate of the control when light-adapted and treated with DCMU, indicating that in these conditions CEF-driven proton pumping may nearly entirely compensate for the loss of LEF in terms of PMF generation. Clearly, the abundance of FNRs in FS1 is able to drive high rates of CEF and the resultant NDH-1 mediated proton pumping.

## 4. Discussion

The FNR isoforms in cyanobacteria, despite having small differences in their kinetics in vitro [17], have very different roles in the regulation of the redox state of the NADPH/Fd pools. It has been shown before that the FS1 has a higher NADP^+^/NADPH ratio and that the mutant does not exhibit a changed PSII/PSI ratio [17], indicating that the FNR_s_ isoform is involved in the reduction of Fd over the reduction of NADPH when it is the primary isoform. This is not the case in the WT, which under normal growth conditions has a larger proportion of FNR_L_ [16], the isoform possessed by the MI6 mutant. The FNR_S_ isoform is maximal during nutrient or light-stress conditions [8]. When the WT was exposed to conditions of high light or low CO_2_, conditions that enhance CEF [42], the FNR_S_ isoform constitutes approximately ¼ of the total amount of FNR in species that can express both isoforms [21]. Presumably, the stress conditions shift the relative demand for the two primary products of the light reactions of photosynthesis: ATP and NADPH. Under stress conditions the demand of ATP increases relative to NADPH, resulting in the observed regulatory modulation in the expression of FNR_S_ [16], which may work in parallel with more direct metabolic control mechanisms, such as the interaction of ATP with the NDH-1 complex [4].

Here, it was shown that CEF is enhanced in FS1, which contains only FNR_S_ (Figure 1E and Figure 2). This was determined by observing the post-illumination rise of chlorophyll fluorescence which has two main components: a fast rise associated with CEF and NDH-1, and a slower rise associated with carbohydrate oxidation [12]. In FS1, a 5 min illumination produces reduction in the PQ pool over the course of illumination, rather than oxidation to a steady-state like in the WT as suggested by the Chl fluorescence transients shown in Figure 1A,E. This is indicative that there are electrons rushing into the PQ pool over the course of illumination in FS1 potentially as a result of recycling of electrons via CEF occurring during illumination such that the rise after the initial oxidation seen in WT and MI6 is either delayed heavily or never reached. However, this interpretation is complicated by the fact that Chl fluorescence yields are simultaneously being modulated by various other processes. These other processes include photoinhibition, orange carotenoid protein quenching, and other regulatory short-term processes, and also appear to be during the decay of fluorescence yields, especially after the M-T rise in *Synechocystis* sp. PCC 6803 [36,37,39,40,41]. After illumination, there is a strong rise in FS1, but not in the WT or MI6 (Figure 1A,C,E), indicative of enhanced electron flow to the PQ pool upon cessation of illumination. When treated with KCN, the PQ pool is generally reduced prior to illumination, yielding a high F_0_, and the PQ pool reduces over the course of illumination, to a point near F_m’_ in the WT, slightly in MI6, and slowly and constantly in FS1 (Figure 1B,D,F). After illumination, each of the strains experiences a strong post-illumination rise at similar rates; however, FS1 had the largest rise of the three strains (Figure 2). To observe the post-illumination rise without inhibition of COX, a short illumination with an MT flash was performed on dark-adapted cells, and again FS1 was shown to have the largest rise in terms of magnitude, and rises at approximately the same rate as the WT (Figure 3, black and green). The MI6 strain had a small shoulder that was quenched well before FS1 or the WT started their PQ pool oxidation (Figure 3, red). These results support the hypothesis established previously [16,17,18] that FNR_S_ contributes more strongly to cyclic electron flow than FNR_L_ does. Because the post-illumination rise is associated with NDH-1 activity [8,14], and because NDH-1 complexes depend on Fd_red_ to power their proton pumping activity [14,22,43], the oxidation state of the NADPH pool during illumination was investigated side-by-side with chlorophyll fluorescence.

The WT and MI6 had similar kinetics of NADPH fluorescence in the absence of KCN, with the WT having overall higher fluorescence during illumination, indicating a slightly more reduced NADPH pool during illumination (Figure 4A,C). Their post-illumination rises, associated with carbohydrate oxidation, are similar in both, being slightly larger in the WT. The FS1 mutant, on the other hand, had strikingly unique kinetics. It had a sharp rise in fluorescence after illumination, shifted ~15 s earlier than the peaks seen in WT and MI6 (Figure 4E). This might be FNR_S_ operating as an NADPH reductase early in illumination due to a large influx of photosynthetically reduced Fd produced by LEF. Because the CBB cycle is regulated by the redox state of the NADPH pool, activating upon heavy reduction [44], this could explain the shifting of the redox state of the NADPH pool by CBB activity to its earlier point in FS1 (Figure 4E). There is steady oxidation of NADPH after this initial peak that is sharper in its fluorescence decline than in the WT or MI6. This is likely due to CBB cycle activity drawing down NADPH, the activity of flavodiiron proteins and COX, as well as Fd reduction by FNR_S_. This oxidation state of the NADPH pool then reaches a steady state where it is consumed as fast as it is produced with a higher NADP^+^/NADPH ratio than the WT or MI6 in the same conditions, matching earlier data [17,18]. Post-illumination fluorescence kinetics of FS1 are also interesting in these conditions, with a large post-illumination fluorescence rise (Figure 4E). It is unlikely that this peak can only be due to increased sugar-oxidation activity due to its magnitude; therefore, FNR_S_ may be acting to dissipate the reduced Fd now that it is no longer being photosynthetically produced. Upon treatment with KCN, the NADPH pool of the WT is almost entirely reduced in the dark, with little increase in fluorescence seen when illuminated (Figure 4B). This is maintained throughout illumination and is followed by a small post-illumination rise. The MI6 has similar kinetics as it had before treatment with KCN; however, fluorescence does not increase as much upon illumination (Figure 4D). It reaches a steady state after the slow oxidation of NADPH by the CBB cycle, and post-illumination experiences a large increase in fluorescence. The FS1 had a fast reduction in NADPH followed by a rapid oxidation, although this was not as rapid as without KCN (Figure 4F). After ~2.5 min of illumination a trough is seen in the fluorescence, and it slowly and steadily rises for the rest of illumination. Due to the lack of COX as an electron sink in the dark, it is possible that the slower oxidation followed by steady reduction in NADPH could be due to a strong reduction in both NADPH and Fd pools in the dark, which would result in a slower exchange of electrons between the two as they become further reduced. The slow rise could then be the FNR_S_ operating to reduce the NADPH pool in conditions where Fd is over-reduced.

The establishment of proton motive force in cyanobacteria is dependent upon photosynthetic electron transport upon illumination. When PSII is inhibited, the major driver of proton pumping by CEF-driven electron transport involving NDH-1 [8]. Because NDH-1 is dependent upon reduced Fd to supply electrons [14,22,43], the ability for the proton gradient to be established in the absence of PSII activity may be indicative of the Fd reduction state. When dark-adapted, the WT has its fastest rate when PSII is active, with the addition of DCMU decreasing the rate by ~50% (Figure 5A and Table 1). FS1 was similar in rates to the WT when DCMU was added, but significantly faster (*p* = 0.01) with only the background inhibitors, including KCN, which yields a more reduced NADPH pool in the dark, potentially providing an increased pool of reductant for Fd reduction (Figure 4E, Figure 6B and Table 1). The rates in MI6 were similar to the WT with only the background inhibitors, and significantly slower than the WT with DCMU added (*p* = 0.01) (Figure 5C and Table 1), achieving only ~47% the rate of the WT when PSII is inactive. When PSII is inhibited, the rate of MI6 is only 27% the rate of MI6 without DCMU, indicating a severe inhibition of CEF in these conditions, likely due to a lack of reduced Fd for consumption. These data indicate that FNR_S_ is an important component in the shifting from dark to light conditions, providing reductant for CEF and contributing to the formation of PMF, even when PSII is active.

Because repeated illumination of the same sample was seen to cause increases in the rate of AO quenching, a pre-illumination of 600 µE m^−2^ s^−1^ over a period of 2 min was applied to the cells, followed by darkness for 4 min, and then measurement of the typical 15 s illuminations at 600 µE m^−2^ s^−1^. This caused rate increases in all the strains measured. The rates in the WT increased by ~1.3× without DCMU and by ~2× with DCMU when pre-illuminated, which when paired with the NADPH fluorescence data, indicates that a heavily reduced NADPH pool is beneficial to CEF-driven PMF generation (Figure 4B and Figure 6A, Table 1). The rate increases in the MI6 mutant were quite dramatic, and without DCMU allowed it to nearly match the other strains. While pre-illumination still improved its rate when treated with DCMU, it remained slower than the other strains (Figure 5C and Table 1). This could be indicative that the FNR_L_ isoform is not able to perform NADPH oxidation in vivo as FNR_S_. The FS1 mutant, had the most dramatic increase in rate with light adaptation, with a DCMU-treated rate 2.7× that in the dark-adapted state with DCMU (Figure 5B and Table 1). When compared to the rate with only the background inhibitors, FS1 was able to achieve ~89% the quenching rate, enough to be not significantly different in rate than the rate without DCMU (*p* = 0.24). These data indicate that the pre-illumination period in the presence of the background inhibitors results in the accumulation of reductant, which may then be consumed to generate PMF. Because the FS1 strain, containing only FNR_S_, has such a dramatically increased rate compared to its counterparts, the FNR_S_ isoform must be providing reduced Fd for consumption by NDH-1 to produce PMF, as it also possesses a heavily oxidized NADPH pool (Figure 4E,F). Because the pre-illumination also increased the rate in MI6 to near WT levels and is the primary FNR isoform present in cells in normal growth conditions, FNR_L_ may be able to provide reduced Fd for CEF and is the likely isoform doing so in the WT. It is, however, limited in its capacity to do so without a period of illumination to adapt to the new redox conditions. This indicates that these isoforms have different functions in maintaining the redox state of the NADP(H)/Fd pools. Because of the improved CEF-driven PMF formation in dark-adapted cells treated with DCMU that the WT possesses over MI6, while being not significantly different from FS1, it is possible that the WT utilizes the small amount of FNR_S_ it possesses in the dark–light transition.

The presence of alternately regulated FNR isoforms in cyanobacteria capable of heterotrophy suggests that they have different roles in their maintenance of the NADPH/Fd redox state (Figure 7). Because the NADPH pool is oxidized over the course of illumination in a mutant expressing only FNR_S_ (Figure 4E), it has enhanced CEF (Figure 2 and Figure 3), and its presence allows for enhanced CEF-driven proton pumping both with and without adaptation to light (Figure 5 and Figure 6, Table 1). This indicates that FNR_S_ is primarily involved in NADPH oxidation/Fd reduction, as reduced Fd is the source of reductant for NDH-1 complexes, the primary driver of proton pumping in the dark–light transition [8]. This also suggests that FNR_S_ is useful in the transition from dark to light, allowing faster adaptation to the light and establishing PMF more quickly to utilize the energy produced. When FNR_L_ is the only isoform present, the establishment of PMF is slower, but upon adaptation to illumination becomes faster in rate of AO quenching to values close to those observed in the WT. The data then suggest that FNR_S_ is useful for allowing for fast adaptation to illumination and likely provides reductant for proton pumping through carbohydrate oxidation in the dark, and FNR_L_ is useful for long-term adaptation to illumination and may be used to reduce Fd for CEF given some time in the light. The roles suggested here also raise a question of whether cyanobacteria that possess both FNR_L_ and FNR_S_ (those capable of heterotrophy) have a more robust response to fluctuating light conditions than their cousins that only possess FNR_L_.

## 5. Conclusions

The FNR isoforms present in cyanobacteria capable of heterotrophy present an interesting and poorly understood system for regulating the redox state of the NADPH/Fd pools. FNR_S_ is utilized in vivo most generally as an NADPH oxidase in the light, allowing for rapid adaptation to the dark–light transition and quick production of a proton gradient. This may allow these bacteria to acclimate more rapidly to altered conditions of photosynthetic electron, flow maintaining redox balance and powering of NDH-1-mediated CEF proton pumping to augment ATP production to match increase demands due to stress. Under typical growth conditions, production of the FNR_L_ isoform is favored. As the FNR_L_ isoform is mostly involved in LEF and the reduction of NADP^+^, it appears to be less effective in reducing Fd even when enough reductant as NADPH is available. The fact that only a subset of cyanobacteria is capable of expressing both isoforms suggests that there is a functional reason why these isoforms are kept in some cyanobacteria, and it perhaps allows these cyanobacteria capable of heterotrophy to adapt more readily to changes in the environment that affect the internal redox pools.

## Figures and Tables

**Figure 1 microorganisms-10-00855-f001:**
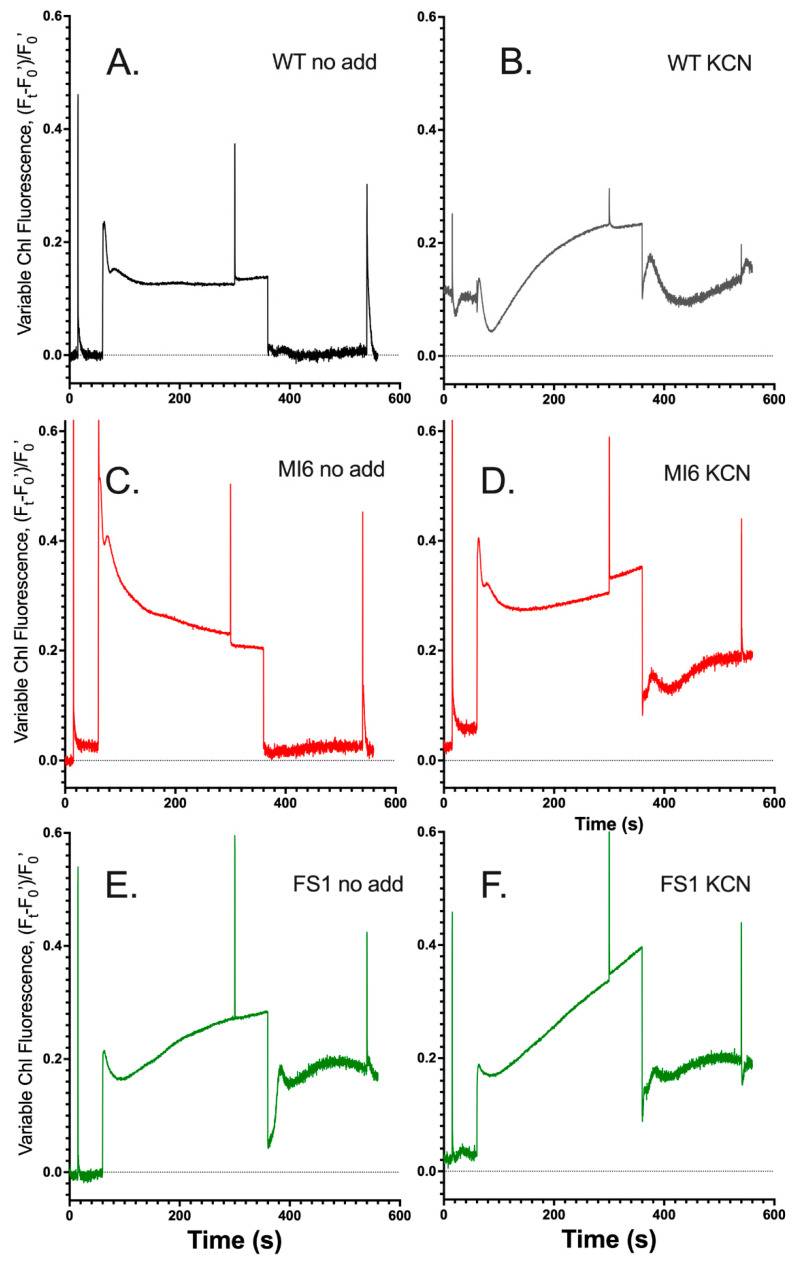
Chlorophyll fluorescence kinetics are distinct in strains lacking either FNR isoform. (**A**): Chlorophyll fluorescence in the DUAL PAM-100 of dark-adapted WT cells in TCK buffer upon illumination for 5 min with 53 µE m^−2^ s^−1^ actinic light with multiple turnover flashes before, during, and after illumination. (**B**): Chlorophyll fluorescence of the WT after treatment with KCN. The trace of the WT with and without KCN is repeated in panels (**C**–**F**) in grey for comparisons. (**C**): MI6 strain. (**D**): MI6 after treatment with KCN. (**E**): Chlorophyll fluorescence of the FS1 strain. (**F**): Chlorophyll fluorescence of FS1 after treatment with KCN.

**Figure 2 microorganisms-10-00855-f002:**
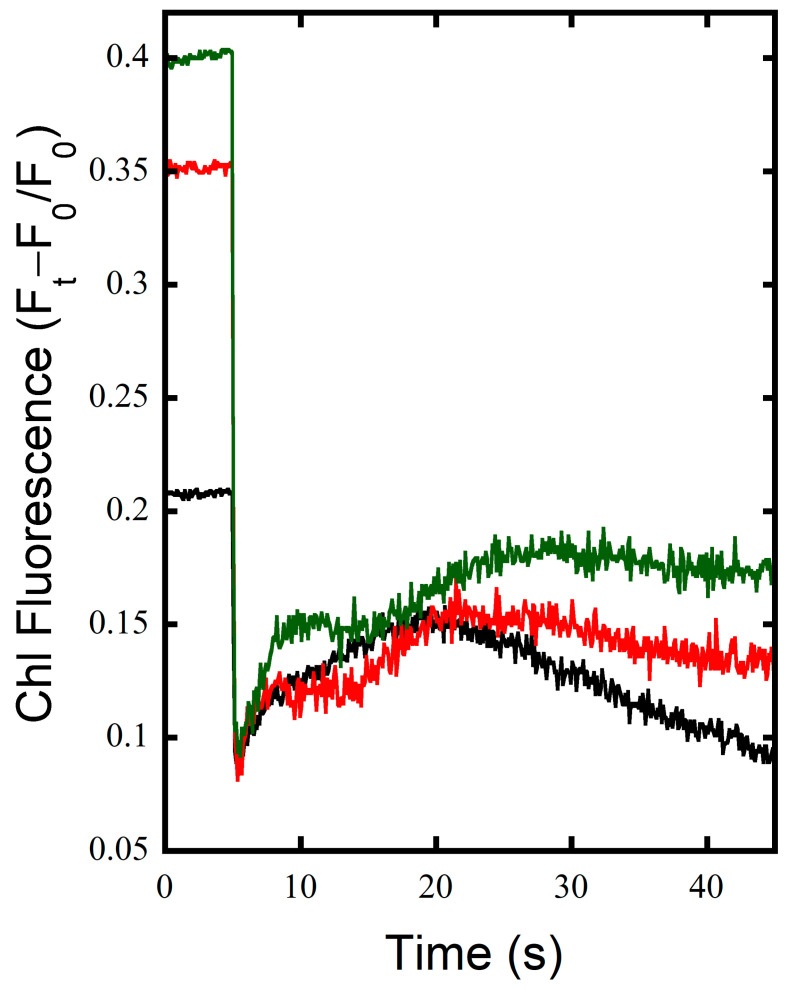
Post-illumination chlorophyll fluorescence rise is enhanced after addition of KCN. Chlorophyll fluorescence in the DUAL PAM-100 of WT (black), MI6 (red), and FS1 (green) cells in TCK buffer with 200 µM KCN upon termination of 5 min illumination with actinic illumination of 53 µE m^−2^ s^−1^.

**Figure 3 microorganisms-10-00855-f003:**
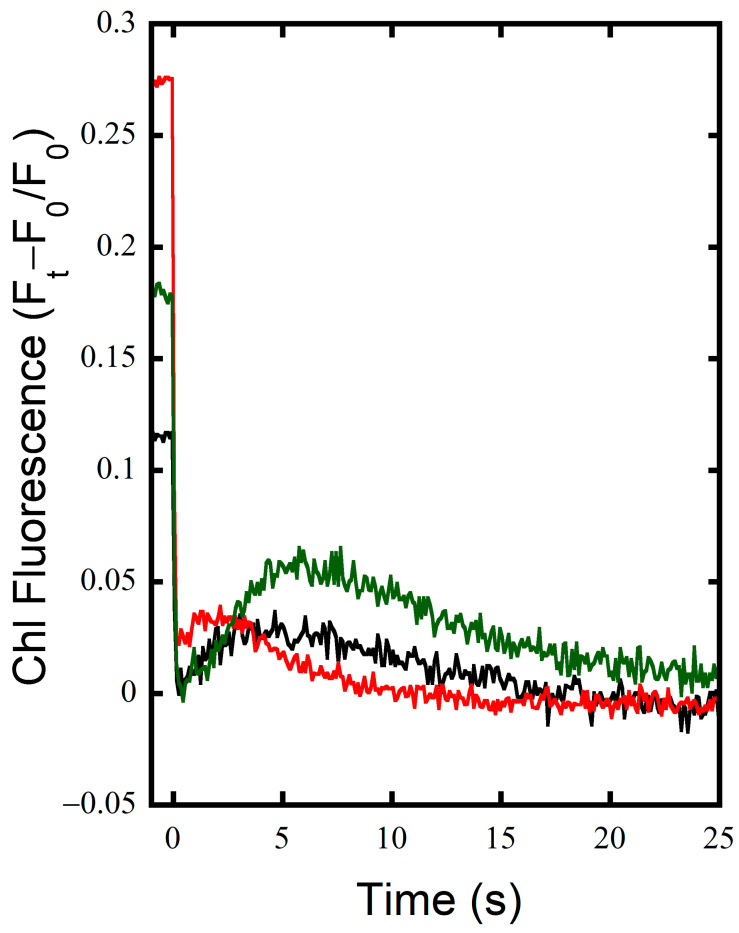
Post-illumination chlorophyll fluorescence rise after short illumination with no inhibitor additions. Chlorophyll fluorescence in the DUAL PAM-100 of dark-adapted WT (black), MI6 (red), and FS1 (green) cells in TCK buffer after actinic illumination with 53 µE m^−2^ s^−1^ for 15 s.

**Figure 4 microorganisms-10-00855-f004:**
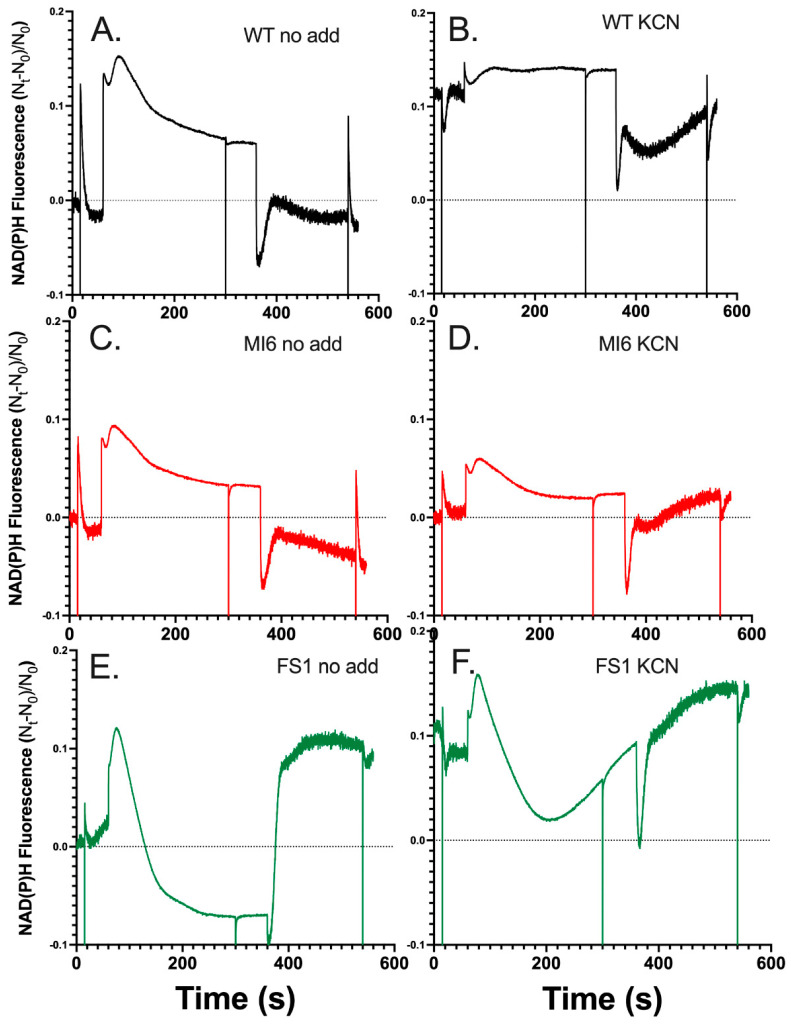
NADPH fluorescence in the WT and strains lacking either FNR isoform. (**A**) NADPH fluorescence in the DUAL PAM-100 of dark-adapted WT cells in TCK buffer upon illumination for 5 min with 53 µE m^−2^ s^−1^ and multiple turnover flashes before, during, and after illumination (cropped here for emphasis). (**B**) NADPH fluorescence of the WT after treatment with KCN. (**C**) NADPH fluorescence of the MI6 strain. (**D**) NADPH fluorescence of MI6 after treatment with KCN. (**E**) NADPH fluorescence of the FS1 strain. (**F**) NADPH fluorescence of FS1 after treatment with KCN.

**Figure 5 microorganisms-10-00855-f005:**
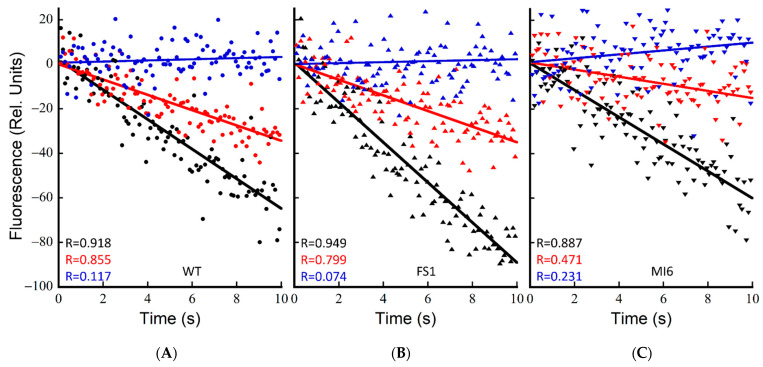
Rates of acidification, *j*_H_^+^, upon illumination of dark-adapted WT, FS1, and MI6. Acridine orange fluorescence of dark-adapted cells in the JTS-100. Cells were dark-adapted with acridine orange (5 µM) for 20 min in TCK buffer. Sample was prepared with Val (10 µM), DCCD (500 µM), and KCN (200 µM) and stirred in the dark for 5 min as the background inhibitors. Actinic illumination (630 nm, 600 μE) was applied for 15 s for measurement. These plots are typical of 3 technical replicates. WT (**A**), FS1 (**B**), and MI6 (**C**) after treatment with the background inhibitors (black symbols), 10 µM DCMU (red symbols), or DCMU and 250 µM CCCP (blue symbols). Data were normalized to the [Chl]/OD of the WT to account for differences in [Chl] per cell.

**Figure 6 microorganisms-10-00855-f006:**
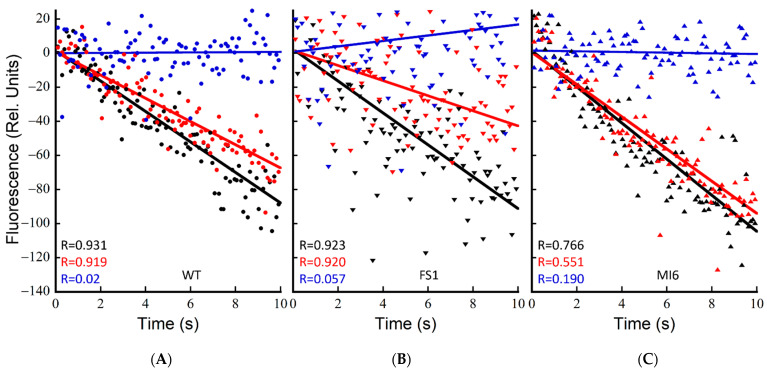
Rates of acidification, *j*_H_^+^, upon illumination of light-adapted WT, FS1, and MI6. Acridine orange fluorescence of dark-adapted cells in the JTS-100. Cells were dark-adapted with acridine orange (5 µM) for 20 min in TCK buffer. Sample was prepared with Val (10 µM), DCCD (500 µM), and KCN (200 µM) and stirred in the dark for 5 min as the background inhibitors. After the addition of inhibitors, cells were pre-illuminated with 600 µE actinic light for 2 min, and cells afforded 4 min in the dark before measurement with 15 s of illumination at the same light intensity. These plots are typical of 3 technical replicates. WT (**A**), FS1 (**B**), and MI6 (**C**) after treatment with the background inhibitors (black symbols), 10 µM DCMU (red symbols), or DCMU and 250 µM CCCP (blue symbols). Data were normalized to the [Chl]/OD of the WT to account for differences in [Chl] per cell.

**Figure 7 microorganisms-10-00855-f007:**
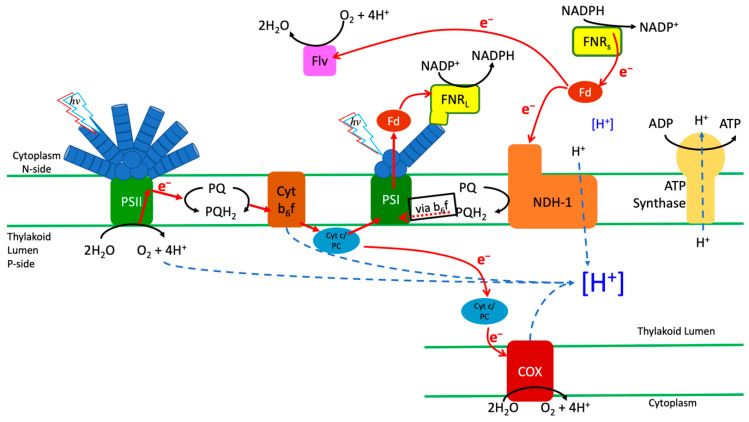
Schematic representation of the electron and proton transport pathways in *Synechocystis* sp. PCC6803 in relation to the large (FNR_L_) and the small soluble (FNR_S_) forms of the FNR protein (yellow). These alternative forms of FNR shift the balance of electron transfer (red arrows) toward either cyclic electron flow or linear electron flow. The mutant FS1, having exclusively FNR_S_, tends to favor electron flow from NADPH to Fd_ox_, which can donate to either the NDH-1 complex or flavodiioron proteins (e.g., Flv2/3). Either of these FNR_S_-favoring pathways can lead to loss of electron from the system by reduction of O_2_ either via cytochrome oxidase (COX) or flavodiioron proteins (Flv). The phycobilisome-attached FNR_L_ appears to be primarily involved in linear electron flow, favoring electron transfer from Fd_red_ produced by PSI to NADPH for CO_2_ fixation.

**Table 1 microorganisms-10-00855-t001:** Relative rates of acidification in dark- and light-adapted states. The rate of acidification, *j*_H_^+^, upon illumination in WT, FS1, and MI6 in the presence of the background inhibitors with either no additional inhibitors, with DCMU, or DCMU and CCCP added. Values are averages of three technical replicates per condition. Data are presented with the standard deviation. Data normalized to [Chl]/OD_750_ of the WT to account for differences in the Chl content per cell in the different strains. Rates are divided into either dark-adapted conditions (top) or light-adapted conditions (bottom). Significance between measured rates are indicated by paired symbols. *p* values: † = 0.0003, * = 0.0003, ‡ = 1 × 10^−8^, ** = 0.0003, ₡ = 0.02, ℓ = 0.004, ◊ = 0.02, • = 6 × 10^−5^, ¤ = 0.005, + = 0.01, ¢ = 5 × 10^−5^, ø = 0.05, § = 0.01, ₠ = 0.007, ж = 0.001.

	Inhibitors Added	Wild-Type	FS1	MI6
Dark-adapted	Val + DCCD + KCN	6.67 ± 0.65 s^−1^ †+	9.00 ± 1.65 s^−1^ ‡+₠	6.08 ± 1.27 s^−1^ ₡•₠
	Val + DCCD + KCN + DCMU	3.44 ± 1.15 s^−1^ *†§	3.52 ± 0.48 s^−1^ **‡ℓ	1.63 ± 1.00 s^−1^ ◊•§ℓ
	Val + DCCD + KCN + DCMU + CCCP	−0.26 ± 0.46 s^−1^	−0.022 ± 0.28 s^−1^	−0.88 ± 0.55 s^−1^
Light-adapted	Val + DCCD + KCN	8.95 ± 2.57 s^−1^	10.60 ± 2.04 s^−1^	9.36 ± 2.59 s^−1^ ₡¤
	Val + DCCD + KCN + DCMU	6.81 ± 0.50 s^−1^ *¢ø	9.45 ± 0.72 s^−1^ **¢ж	4.43 ± 2.22 s^−1^ ◊¤øж
	Val + DCCD + KCN + DCMU + CCCP	−0.08 ± 0.97 s^−1^	0.21 ± 0.27 s^−1^	−1.61 ± 1.67 s^−1^

## Data Availability

Not applicable.

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
