# Peer review of "Cyclic Electron Flow-Coupled Proton Pumping in Synechocystis sp. PCC6803 Is Dependent upon NADPH Oxidation by the Soluble Isoform of Ferredoxin:NADP-Oxidoreductase"

_microorganisms, 2022, doi:10.3390/microorganisms10050855_

Round 1

Reviewer 1 Report

General comments and suggestions on the manuscript microorganisms-1641749:

I consider the manuscript well organized, and with sufficient and valid experimental data to sustain their conclusions, and good quality figures and Tables. However, there are some unclear phrases in manuscript that need revision (see the annotated MS, and the list below, with specific comments and suggestions). Also, I consider that an additional schematic figure representing different alternative cyclic electron transport paths in Synechocystis sp. PCC 6803 would be helpful for the understanding of their analysis of experimental data. Also, if acceptable, I suggest to the authors to also interpret their data in the light of state transitions, as the causes of Chl fluorescence decrease after the S-M rise are questioned in some relatively recent papers on this topic (see e.g., Bernát et al. 2018; Stirbet et al. 2019).  

Bernát G, Steinbach G, Kaňa R, Govindjee, Misra AN, Prašil O (2018) On the origin of the slow M-T chlorophyll a fluorescence decline in cyanobacteria: interplay of short-term light-responses. Photosynth Res 136(2):183-198

Stirbet A, Lazar D, Papageorgiou CG, Govindjee. 2019. Chlorophyll a fluorescence in cyanobacteria: relation to photosynthesis. In: Mishra AK, Tiwari DN, Rai AN. eds. Cyanobacteria – from basic science to applications. London: Academic Press, 79–130.   

List of specific comments and suggestions

ABSTRACT

  • P1, L23: At the beginning of the line, replace “The” with “We found that the”. This way the reader can clearly see that these are the results obtained by the authors.
  1. INTRODUCTION
  • P1, L35: Besides the references [1] and [2], I suggest the additional citation of one or more recent reviews on CEF. See e.g.:

Nawrocki WJ, Bailleul B, Picot D, Cardol P, Rappaport F, Wollman F-A, Joliot P (2019) The mechanism of cyclic electron flow. BBA-Bioenergetics 1860: 433-438

Nikkanen L, Solymosi D, Jokel M, Allahverdiyeva Y (2021) Regulatory electron transport pathways of photosynthesis in cyanobacteria and microalgae: Recent advances and biotechnological prospects. Physiologia Plantarum 173: 514-525

  • P2, L46: Replace “net production” with “net consumption”. A net NADPH consumption would affect negatively the carbon assimilation rate. So, I think that it is preferable to write "net consumption of NADPH", rather than net production.
  • P2, L47: In the phrase “The fraction of electrons recycled through CEF relative to LEF provides a mechanism for cells to adjust the output ratio of NADPH and ATP to accommodate different metabolic demands and fluctuating environmental conditions”, I suggest to: (1) replace “provides a mechanism for” with “allows the”; (2) replace “of NADPH and ATP” with “ATP/NADPH”.
  • P2, L54-57: Relative to the phrase “Although alternative cyclic paths involving different membrane complexes have been discovered, in all cases electrons are transferred to the plastoquinone (PQ) pool, thereby re-entering the electron transport chain to be further utilized to produce NADPH and Fdred via PSI and subsequently recycled via CEF”. I suggest to cite key references on alternative cyclic paths. Also, I think that a schematic figure of different alternative cyclic electron transport paths in Synechocystis PCC 6803 would be very helpful for the reader.
  • P2, L61: Replace “difference kinetics apparent” with “different apparent kinetics”.
  • P2, L62-64: I suggest to replace the phrase “One of the mechanisms by which the redox state of the NADPH/Fd pool is maintained in these organisms is by the presence of Ferredoxin:NADP-oxidoreductase (FNR)” with “One way to maintain the redox state of the NADPH/Fd pool in these organisms is through the redox activity of Ferredoxin:NADP-oxidoreductase (FNR).”
  • P2, L64-66: In the phrase between the lines 64-66, I suggest to: (1) Add “two-electron reduction of” after “photosynthetic”; (2) Replace “reduction from” with “to NADPH by Fdred produced on”.
  • P2, L68-72: Rewrite the text between the lines 68-72, and use short and clear sentences.
  • P2, L74: After “is”, add “soluble and”.
  • P2, L84: Delete “conditions” before “which”.
  • P2, L86: Replace “mechanisms” with “apparatus”.
  1. MATERIALS AND METHODS
  • P3, L98-101: Rewrite the text between the lines 98-101, and use short and clear sentences.
  • P3, L108: Replace “yield” with “kinetics”.
  • P3, L109: Write the wavelengths of the actinic and measuring lights used during the measurements. These are relevant information in Chl fluorescence measurements.
  • P3, L108: Replace “yield” with “kinetics”.
  • P3, L119: Add “measurements” after “Chl fluorescence”
  1. RESULTS

3.1. FNRs has a large contribution to NDH-1 cyclic electron flow

  • P3, L140: Important older paper on post-illumination Chl fluorescence kinetics may be also added to the references [1,5,7,25]. Like e.g.:

Mi H, Endo T, Ogawa T, Asada K (1995) Thylakoid membrane-bound pyridine nucleotide dehydrogenase complex mediates cyclic electron transport in the cyanobacteria Synechocystis PCC 6803. Plant Cell Physiol 36: 661–668

Tanaka Y, Katada S, Ishikawa H, Ogawa T, Takabe T (1997) Electron flow front NAD(P)H dehydrogenase to photosystem I is required for adaptation to salt shock in the cyanobacterium Synechocystis sp. PCC6803. Plant Cell Physiol 38:1311–1318

  • P4, L149: If appropriate, add “more” before “reduced”.
  • P4, L156-159: Please check the text between the lines 156-158. It was affirmed earlier (line 87) that FNRs operates mainly as a soluble NADPH oxidase, when its activity would lead to Fd reduction, not Fdred oxidation.
  • P4, L172: Cite here, and add to the reference list, the important paper on Chl fluorescence kinetics during post-illumination: Shikanai T, Endo T, Hashimoto T, Yamada Y, Asada K, Yokota A (1998) Directed disruption of the tobacco ndhB gene impairs cyclic electron flow around photosystem I. Proc Natl Acad Sci U S A 95(16): 9705-9709.
  • P4, L177: (1) Replace “to” with “, so that the Chl fluorescence increases”; (2) add “to the” before “Fm”; (3) Replace “levels” with “level”.
  • P4, L182: Add “fluorescence” before “rise”.
  • P4, L187: Add “sec” after “12-30”.
  • P4, L192: Verify. See my comment at line 157.
  • P6, L211: Add “actinic light” before “with”.

3.2. FNRS enhances NADPH oxidation during illumination

  • P7, L238: Besides the references 27-29, I suggest to add the one shown below, in which measurements on WT Synechocystis PCC 6803 and its RpaC mutant (locked in state 1) confirmed that the slow S to M fluorescence rise is due to a State-2 to State-1 transition: Kaňa R, Kotabová E, Komárek O, Šedivá B, Papageorgiou GC, Govindjee G, Prášil O (2012) The slow S to M fluorescence rise in cyanobacteria is due to a state 2 to state 1 transition. Biochim Biophys Acta 1817: 1237–1247
  • P8, L271: (1) Add “At the beginning of” before “Post-illumination”, and change “P” with “p”; (2) Add comma after “Post-illumination”; (3) Replace “it has a” with “the NADPH fluorescence”; (4) Add “in MI6 had a” before “greater; (5) Delete “in”; (6) Replace “has” with a comma’
  • P8, L278: (1) Delete “to a”; (2) Replace “large” with “larger”; (3) Delete “magnitude”
  • P8, L280: Add “on NADPH fluorescence” after “data”
  • P8, L284: (1) Add “reported” before “ratio”; Add comma after “FS1”.
  • P8, L285: Add “NADPH pool” after “oxidized”.
  • P9, L296: Where it is the gray WT trace in the panels C-F in the Fig. 4?

3.3. The presence of FNRs enhances NDH-1 powered proton pumping

  • P10, L325: Add comma after “control”
  1. DISCUSSION
  • P12, L408-409: Note: PQ pool reduction in FS1 and oxidation in WT during 5min illumination are correct. However, Chl fluorescence transients shown in the Figs 1A,E are also influenced by modifications in PSII absorption cross-section associated with state transitions taking place in these samples. Maybe, these should be also acknowledged in the discussion of the data here.
  • P12, L410: Add “as suggested by the Chl fluorescence transients shown in” before “Fig. 1A,E”.
  • P13, L444: Replace comma after “Post-illumination” with “fluorescence kinetics of”.
  • P13, L448: (1) Replace “WT” with “NADPH”; Add “in WT” after ”pool”.
  • P13, L460: (1) Add “the” before “NADPH”; (2) Add “pool” after “NADPH”.
  • P14, L472: (1) Replace comma with point after “1)”; (2) Replace “while the” with “The”.
  • P14, L507: Replace “depends some on” with “uses for this”.
  • P14, L519: (1) Replace “slowed” with “slower”; (2) Replace “it is only sped up” with “but becomes faster”.
  • P15, L520: (1) Replace “and upon adaptation is close to WT” with “when the”; (2) Add “are closer to those observed in WT” after “quenching”.

Figure S1

  • P18, L641-642: Rewrite the first phrase between the lines 641-642, because it makes no sense.

Suggested minor changes:

  • P1, L13: Add a comma after “transfer”.
  • P1, L21: Add a comma after “activity”.
  • P1, L37: (“In plants, algae, and cyanobacteria the reducing power produced by the photosynthetic light reactions is stored in the form of NADPH and Fdred, which is then consumed by anabolic processes, mainly CO2 fixation via the Calvin-Benson-Bassham (CBB) cycle.”) I suggest to replace “which is then” with “with the former being”.
  • P2, L47: Add a comma after “chloroplasts”.
  • P2, L59: Replace “direct” with “directly”.
  • P2, L67: After “proximity”, add “of”.
  • P2, L90: Add a comma after “[7, 14]”.
  • P3, L142: (1) Replace “a” with “the” before “5 minute”; (2) Delete “the” before “WT”.
  • P3, L143: (1) Replace “have” with “had”; (2) add “fluorescence” before “kinetics”.
  • P3, L144: Replace “is” with “was”.
  • P3, L145: (1) Add “which” before “potentially”; (2) Replace “obscuring” with “obscured”
  • P3, L146: Replace “maintains” with “maintained”.
  • P3, L146: Replace “shows” with “showed”.
  • P4, L150: (1) Replace “exhibits” with “exhibited”; (2) Add comma after “yield”; (3) Replace “staruating” with “saturating”.
  • P4, L151: Add comma after “period”.
  • P4, L153: Add comma after “complex”.
  • P4, L179: Add comma after “illumination”.
  • P4, L188: To avoid redundance, replace “occurs” with “takes place”.
  • P4, L189: Add comma after “FS1”.
  • P4, L189: Add comma after “illumination”.
  • P4, L194: Add “Chl fluorescence” before “rise”.
  • P4, L197: Replace “tha” with “the”.
  • P4, L202: (1) Replace comma after “(Fig. 3) with point; Replace “indicating” with “These results indicate”.
  • P6, L221: Replace “at” with “of”.
  • P8, L270: 2 in CO2, should be written as subscript
  • P8, L272: (1) Replace “continues” with “continued”; (2) Replace “has” with “had”.
  • P8, L274: Replace “has” with “had”.
  • P8, L277: Replace “is” with “was”.
  • P12, L401: (1) Add comma after “increases”; (2) replace “relative” with “comparative”; (3) Add comma after “NADPH”.
  • P12, L403: Add comma after “mechanisms”.
  • P13, L414: Add comma after “illumination”.
  • P13, L425: For consistency, write "Fdred" with "red" as subscript, not superscript.
  • P13, L437: Replace “of” with “to”.
  • P14, L481: Add comma after “4min”.
  • P14, L486: Add comma after “dramatic”.
  • P14, L500: Delete “, did so”.
  • P14, L501: (1) Add “be” after “may”; (2) Add comma after “CEF”.
  • P14, L502: Replace comma with semicolon.
  • P14, L506: Add comma after “MI6”.
  • P14, L514: Replace “suggests” with “indicates”.
  • P15, L520: Add comma after “illumination”.
  • P15, L523: Replace “and” with “, but”.
  • P15, L524: Add “some” before “time”.

Author Response

Many thanks for the detailed critique and suggestions including some interesting papers that we should have been aware of and only now incorporate the ideas into the revised manuscript.

Reviewer 1

I consider the manuscript well organized, and with sufficient and valid experimental data to sustain their conclusions, and good quality figures and Tables. However, there are some unclear phrases in manuscript that need revision (see the annotated MS, and the list below, with specific comments and suggestions). Also, I consider that an additional schematic figure representing different alternative cyclic electron transport paths in Synechocystis sp. PCC 6803 would be helpful for the understanding of their analysis of experimental data. Also, if acceptable, I suggest to the authors to also interpret their data in the light of state transitions, as the causes of Chl fluorescence decrease after the S-M rise are questioned in some relatively recent papers on this topic (see e.g., Bernát et al. 2018; Stirbet et al. 2019). 

Response:

Thank-you for these excellent comments, although we are not sure whether we can confidently fully address them, we have taken these aspects into account. We have gone through the manuscript again and addressed the unclear text.  The comments regarding state transitions and the useful papers on the fluorescence yield decrease after the S-M rise has prompted us to attempt to improve the interpretations of the observations along with inclusion of citations.

ABSTRACT

 P1, L23: At the beginning of the line, replace “The” with “We found that the”. This way the reader can clearly see that these are the results obtained by the authors.

Response: Thank you for the response, it has been included.

    INTRODUCTION

 P1, L35: Besides the references [1] and [2], I suggest the additional citation of one or more recent reviews on CEF. See e.g.:

Nawrocki WJ, Bailleul B, Picot D, Cardol P, Rappaport F, Wollman F-A, Joliot P (2019) The mechanism of cyclic electron flow. BBA-Bioenergetics 1860: 433-438

Nikkanen L, Solymosi D, Jokel M, Allahverdiyeva Y (2021) Regulatory electron transport pathways of photosynthesis in cyanobacteria and microalgae: Recent advances and biotechnological prospects. Physiologia Plantarum 173: 514-525

Response: Thank you for the response, it has been included.

 P2, L46: Replace “net production” with “net consumption”. A net NADPH consumption would affect negatively the carbon assimilation rate. So, I think that it is preferable to write "net consumption of NADPH", rather than net production.

Response: Thank you for the response, it has been included.

 P2, L47: In the phrase “The fraction of electrons recycled through CEF relative to LEF provides a mechanism for cells to adjust the output ratio of NADPH and ATP to accommodate different metabolic demands and fluctuating environmental conditions”, I suggest to: (1) replace “provides a mechanism for” with “allows the”; (2) replace “of NADPH and ATP” with “ATP/NADPH”.

Response: Thank you for the response, it has been included.

 P2, L54-57: Relative to the phrase “Although alternative cyclic paths involving different membrane complexes have been discovered, in all cases electrons are transferred to the plastoquinone (PQ) pool, thereby re-entering the electron transport chain to be further utilized to produce NADPH and Fdred via PSI and subsequently recycled via CEF”. I suggest to cite key references on alternative cyclic paths. Also, I think that a schematic figure of different alternative cyclic electron transport paths in Synechocystis PCC 6803 would be very helpful for the reader.

Response: Thank you for the response, we have addressed this in a section added to the introduction in lines 58-65, as well as the inclusion of a CEF schematic figure.

 P2, L61: Replace “difference kinetics apparent” with “different apparent kinetics”.

Response: Thank you for the response, it has been included.

 P2, L62-64: I suggest to replace the phrase “One of the mechanisms by which the redox state of the NADPH/Fd pool is maintained in these organisms is by the presence of Ferredoxin:NADP-oxidoreductase (FNR)” with “One way to maintain the redox state of the NADPH/Fd pool in these organisms is through the redox activity of Ferredoxin:NADP-oxidoreductase (FNR).”

Response: Thank you for the response, it has been included.

 P2, L64-66: In the phrase between the lines 64-66, I suggest to: (1) Add “two-electron reduction of” after “photosynthetic”; (2) Replace “reduction from” with “to NADPH by Fdred produced on”.

Response: Thank you for the response, it has been included.

 P2, L68-72: Rewrite the text between the lines 68-72, and use short and clear sentences.

Response: Thank you for the response, it has been rewritten.

 P2, L74: After “is”, add “soluble and”.

Response: Thank you for the response, it has been included.

 P2, L84: Delete “conditions” before “which”.

Response: Thank you for the response, it has been included.

 P2, L86: Replace “mechanisms” with “apparatus”.

Response: Thank you for the response, it has been included.

    MATERIALS AND METHODS

 P3, L98-101: Rewrite the text between the lines 98-101, and use short and clear sentences.

Response: Thank you for the response, it has been included.

 P3, L108: Replace “yield” with “kinetics”.                                  

Response: Thank you for the response, it has been included.

 P3, L109: Write the wavelengths of the actinic and measuring lights used during the measurements. These are relevant information in Chl fluorescence measurements.

Response: Thank you for the response, the wavelengths have been included.

 P3, L108: Replace “yield” with “kinetics”.

Response: Thank you for the response, it has been included.

 P3, L119: Add “measurements” after “Chl fluorescence”

Response: Thank you for the response, it has been included.

    RESULTS

3.1. FNRs has a large contribution to NDH-1 cyclic electron flow

 P3, L140: Important older paper on post-illumination Chl fluorescence kinetics may be also added to the references [1,5,7,25]. Like e.g.:

Mi H, Endo T, Ogawa T, Asada K (1995) Thylakoid membrane-bound pyridine nucleotide dehydrogenase complex mediates cyclic electron transport in the cyanobacteria Synechocystis PCC 6803. Plant Cell Physiol 36: 661–668

Tanaka Y, Katada S, Ishikawa H, Ogawa T, Takabe T (1997) Electron flow front NAD(P)H dehydrogenase to photosystem I is required for adaptation to salt shock in the cyanobacterium Synechocystis sp. PCC6803. Plant Cell Physiol 38:1311–1318

Response: Thank you for the response, it has been included.

 P4, L149: If appropriate, add “more” before “reduced”.

Response: Thank you for the response, it has been included.

 P4, L156-159: Please check the text between the lines 156-158. It was affirmed earlier (line 87) that FNRs operates mainly as a soluble NADPH oxidase, when its activity would lead to Fd reduction, not Fdred oxidation.

Response: Thank you for the response, this issue has been fixed.

 P4, L172: Cite here, and add to the reference list, the important paper on Chl fluorescence kinetics during post-illumination: Shikanai T, Endo T, Hashimoto T, Yamada Y, Asada K, Yokota A (1998) Directed disruption of the tobacco ndhB gene impairs cyclic electron flow around photosystem I. Proc Natl Acad Sci U S A 95(16): 9705-9709.

Response: Thank you for the response, it has been included.

 P4, L177: (1) Replace “to” with “, so that the Chl fluorescence increases”; (2) add “to the” before “Fm”; (3) Replace “levels” with “level”.

Response: Thank you for the response, it has been included.

 P4, L182: Add “fluorescence” before “rise”.

Response: Thank you for the response, it has been included.

 P4, L187: Add “sec” after “12-30”.

Response: Thank you for the response, it has been included.

 P4, L192: Verify. See my comment at line 157.

Response: Thank you for the response, this has been fixed.

 P6, L211: Add “actinic light” before “with”.

Response: Thank you for the response, it has been included.

3.2. FNRS enhances NADPH oxidation during illumination

 P7, L238: Besides the references 27-29, I suggest to add the one shown below, in which measurements on WT Synechocystis PCC 6803 and its RpaC− mutant (locked in state 1) confirmed that the slow S to M fluorescence rise is due to a State-2 to State-1 transition: Kaňa R, Kotabová E, Komárek O, Šedivá B, Papageorgiou GC, Govindjee G, Prášil O (2012) The slow S to M fluorescence rise in cyanobacteria is due to a state 2 to state 1 transition. Biochim Biophys Acta 1817: 1237–1247

Response: Thanks--indeed, good point and we use these important references to qualify the interpretations and connect these observations the cited earlier findings.  Also, in the discussion of the Chl fluorescence traces now.

 P8, L271: (1) Add “At the beginning of” before “Post-illumination”, and change “P” with “p”; (2) Add comma after “Post-illumination”; (3) Replace “it has a” with “the NADPH fluorescence”; (4) Add “in MI6 had a” before “greater; (5) Delete “in”; (6) Replace “has” with a comma’

Response: Thank you for the response, it has been included.

 P8, L278: (1) Delete “to a”; (2) Replace “large” with “larger”; (3) Delete “magnitude”

Response: Thank you for the response, it has been included.

 P8, L280: Add “on NADPH fluorescence” after “data”

Response: Thank you for the response, it has been included.

 P8, L284: (1) Add “reported” before “ratio”; Add comma after “FS1”.

Response: Thank you for the response, it has been included.

 P8, L285: Add “NADPH pool” after “oxidized”.

Response: Thank you for the response, it has been included.

 P9, L296: Where it is the gray WT trace in the panels C-F in the Fig. 4?

Response: Thank you for the response, it has been included.

3.3. The presence of FNRs enhances NDH-1 powered proton pumping

 P10, L325: Add comma after “control”

Response: Thank you for the response, I opted not to include the comma to avoid breaking up the phrase.

    DISCUSSION

 P12, L408-409: Note: PQ pool reduction in FS1 and oxidation in WT during 5min illumination are correct. However, Chl fluorescence transients shown in the Figs 1A,E are also influenced by modifications in PSII absorption cross-section associated with state transitions taking place in these samples. Maybe, these should be also acknowledged in the discussion of the data here.

Response:

 P12, L410: Add “as suggested by the Chl fluorescence transients shown in” before “Fig. 1A,E”.

 P13, L444: Replace comma after “Post-illumination” with “fluorescence kinetics of”.

 P13, L448: (1) Replace “WT” with “NADPH”; Add “in WT” after ”pool”.

 P13, L460: (1) Add “the” before “NADPH”; (2) Add “pool” after “NADPH”.

 P14, L472: (1) Replace comma with point after “1)”; (2) Replace “while the” with “The”.

 P14, L507: Replace “depends some on” with “uses for this”.

 P14, L519: (1) Replace “slowed” with “slower”; (2) Replace “it is only sped up” with “but becomes faster”.

 P15, L520: (1) Replace “and upon adaptation is close to WT” with “when the”; (2) Add “are closer to those observed in WT” after “quenching”.

Response: Thank you for the corrections, which have now been integrated and comma placement considered.

Figure S1

 P18, L641-642: Rewrite the first phrase between the lines 641-642, because it makes no sense.

Response: Thank you for pointing this out, this has been rewritten.

Suggested minor changes:

 P1, L13: Add a comma after “transfer”.

 P1, L21: Add a comma after “activity”.

 P1, L37: (“In plants, algae, and cyanobacteria the reducing power produced by the photosynthetic light reactions is stored in the form of NADPH and Fdred, which is then consumed by anabolic processes, mainly CO2 fixation via the Calvin-Benson-Bassham (CBB) cycle.”) I suggest to replace “which is then” with “with the former being”.

 P2, L47: Add a comma after “chloroplasts”.

 P2, L59: Replace “direct” with “directly”.

 P2, L67: After “proximity”, add “of”.

 P2, L90: Add a comma after “[7, 14]”.

 P3, L142: (1) Replace “a” with “the” before “5 minute”; (2) Delete “the” before “WT”.

 P3, L143: (1) Replace “have” with “had”; (2) add “fluorescence” before “kinetics”.

 P3, L144: Replace “is” with “was”.

 P3, L145: (1) Add “which” before “potentially”; (2) Replace “obscuring” with “obscured”

 P3, L146: Replace “maintains” with “maintained”.

 P3, L146: Replace “shows” with “showed”.

 P4, L150: (1) Replace “exhibits” with “exhibited”; (2) Add comma after “yield”; (3) Replace “staruating” with “saturating”.

 P4, L151: Add comma after “period”.

 P4, L153: Add comma after “complex”.

 P4, L179: Add comma after “illumination”.

 P4, L188: To avoid redundance, replace “occurs” with “takes place”.

 P4, L189: Add comma after “FS1”.

 P4, L189: Add comma after “illumination”.

 P4, L194: Add “Chl fluorescence” before “rise”.

 P4, L197: Replace “tha” with “the”.

 P4, L202: (1) Replace comma after “(Fig. 3) with point; Replace “indicating” with “These results indicate”.

 P6, L221: Replace “at” with “of”.

 P8, L270: 2 in CO2, should be written as subscript

 P8, L272: (1) Replace “continues” with “continued”; (2) Replace “has” with “had”.

 P8, L274: Replace “has” with “had”.

 P8, L277: Replace “is” with “was”.

 P12, L401: (1) Add comma after “increases”; (2) replace “relative” with “comparative”; (3) Add comma after “NADPH”.

 P12, L403: Add comma after “mechanisms”.

 P13, L414: Add comma after “illumination”.

 P13, L425: For consistency, write "Fdred" with "red" as subscript, not superscript.

 P13, L437: Replace “of” with “to”.

 P14, L481: Add comma after “4min”.

 P14, L486: Add comma after “dramatic”.

 P14, L500: Delete “, did so”.

 P14, L501: (1) Add “be” after “may”; (2) Add comma after “CEF”.

 P14, L502: Replace comma with semicolon.

 P14, L506: Add comma after “MI6”.

 P14, L514: Replace “suggests” with “indicates”.

 P15, L520: Add comma after “illumination”.

 P15, L523: Replace “and” with “, but”.

 P15, L524: Add “some” before “time”.

Response: Thank you for the correction, these have been integrated.

Reviewer 2 Report

The submitted manuscript investigates the roles of two forms of FNR in cyclic electron flow. Previous work (ref 9) has concluded that FNRS is involved in NADPH oxidation and FNRL in NADPH reduction. Hence a clearer explanation of the gap in knowledge that this study is addressing would be welcome. The conclusions rely heavily on the interpretation of fluorescence data from whole cells which is not an easy task. Overall, a paper for the specialists.

Points

  1. Line 19: what are the optimal growth conditions?
  2. The authors seem to overlook in the Introduction the importance of Flv1/3 and pseudocyclic electron flow in generating ATP at the expense of NADPH. There is also no mention of Pgr5 and little discussion of respiratory complexes.
  3. Fig 4 in ref 9 suggests that there is upregulation of FNRs in the FS1 mutant and small amount of cleavage product in the M16 mutant. This raises the possibility that levels of total FNR activity might vary in the mutants and be dependent on growth conditions and that soluble FNR might accumulate in M16 due to proteolytic cleavage. To address this, activity assays could be done plus immunoblotting experiments similar to that in ref 9.
  4. Line 112: more detail on the artefact is needed. The original data and corrected data should be presented.
  5. The use of chlorophyll fluorescence to study PSII activity and reduction state of the PQ pool is notoriously difficult as discussed by Santabarbara for instance. How do the authors tease out changes to fluorescence because of state transitions versus reduction of PSII because of enhance reduction state of the PQ pool? Normally, fluorescence changes in response to a pulse of far-red light to oxidize the PQ pool are used to assess PQ reduction state. Have the authors used this approach? Without this, or an alternative approach, it is difficult to draw concrete conclusions from the chlorophyll fluorescence data
  6. Have the authors looked at the re-reduction of P700+ as a complementary approach to assess impact on cyclic electron flow?
  7. Line 308: a contribution of a sodium gradient to pmf needs more explanation.
  8. What are the photoautotrophic growth rates of the two mutants compared to WT. If FNRs is used primarily for NADPH oxidation then one would expect a dramatic impact on growth rate.

Author Response

Many thanks for the detailed and very useful critique and suggestions. We agree with virtually every comment and have tried to incorporate these into this form of the manuscript adding improvements where possible and qualifying interpretations based upon the ideas given in the critique.

"The submitted manuscript investigates the roles of two forms of FNR in cyclic electron flow. Previous work (ref 9) has concluded that FNRS is involved in NADPH oxidation and FNRL in NADPH reduction. Hence a clearer explanation of the gap in knowledge that this study is addressing would be welcome. The conclusions rely heavily on the interpretation of fluorescence data from whole cells which is not an easy task. Overall, a paper for the specialists.

Response:

Points

  1. Line 19: what are the optimal growth conditions?

Response: Thank you for the response, “optimal” has been changed to “typical.”

  1. The authors seem to overlook in the Introduction the importance of Flv1/3 and pseudocyclic electron flow in generating ATP at the expense of NADPH. There is also no mention of Pgr5 and little discussion of respiratory complexes.

Response: Thank you for this important comment and oversight on our part.  This is important both for background and possibly interpreting the results.  We have added a brief section discussing Pgr5 and Flv1/3 in the introduction lines 59-65.  Also, we now discuss decreases in NADP(H) levels under illumination in the strain only having the small form (FS1) as being oxidized by pathways potentially including Flv1/3. 

  1. Fig 4 in ref 9 suggests that there is upregulation of FNRs in the FS1 mutant and small amount of cleavage product in the M16 mutant. This raises the possibility that levels of total FNR activity might vary in the mutants and be dependent on growth conditions and that soluble FNR might accumulate in M16 due to proteolytic cleavage. To address this, activity assays could be done plus immunoblotting experiments similar to that in ref 9.

Response: Thank you for the response, a small section discussing this has been included in lines 94-97

  1. Line 112: more detail on the artefact is needed. The original data and corrected data should be presented.

Response: Thank you for the response, a small section discussing this has been included in lines 130-134

  1. The use of chlorophyll fluorescence to study PSII activity and reduction state of the PQ pool is notoriously difficult as discussed by Santabarbara for instance. How do the authors tease out changes to fluorescence because of state transitions versus reduction of PSII because of enhance reduction state of the PQ pool? Normally, fluorescence changes in response to a pulse of far-red light to oxidize the PQ pool are used to assess PQ reduction state. Have the authors used this approach? Without this, or an alternative approach, it is difficult to draw concrete conclusions from the chlorophyll fluorescence data.

Response: Thank-you for this important critical comment. Unfortunately, we did not utilize this approach, but now qualify the conclusions highlighting some of the uncertainties associated with state-changes and other quenching processes in both the Results and Discussion sections. Additionally, we are relying more heavily on the post-illumination fluorescence yield rise differences to support the conclusions that the FNRS form enhances cyclic electron flow.

  1. Have the authors looked at the re-reduction of P700+ as a complementary approach to assess impact on cyclic electron flow?

Response: Thank you for the response, we did not utilize P700+ measurements and opted to use the post-illumination fluorescence rise instead to observe the phenomenological changes that are observable between the mutants and to compare them to ∆pH changes in the dark-light transition. Based on our earlier work, we utilized mutants with known CEF impairments and observed that their ∆pH formation was also impaired. Here, we intended to observe whether CEF is enhanced in general by observation of the post-illumination fluorescence rise magnitude and determine whether CEF-driven proton pumping is also enhanced in these mutants expressing different isoforms of FNR.

  1. Line 308: a contribution of a sodium gradient to pmf needs more explanation.

Response: Thank you for the response, we included a brief statement about the contribution of Na+ gradient to pmf in lines 336-339.

  1. What are the photoautotrophic growth rates of the two mutants compared to WT. If FNRs is used primarily for NADPH oxidation then one would expect a dramatic impact on growth rate.

Response: Thank you for the response, this has been addressed in a small section in the introduction.